# Pathogenic *LRRK2* mutations cause loss of primary cilia and Neurturin in striatal parvalbumin interneurons

Yu-En Lin[1,2] , Ebsy Jaimon[1,2] , Francesca Tonelli[2,3] , Suzanne R Pfeffer[1,2]

**Parkinson's disease–associated, activating mutations in the LRRK2 kinase block primary cilium formation in cell culture and in specific cell types in the brain. In the striatum that is important for movement control, about half of astrocytes and cholinergic interneurons, but not the predominant medium spiny neurons, lose their primary cilia. Here, we show that mouse and human striatal parvalbumin interneurons that are inhibitory regulators of movement also lose primary cilia. Without cilia, these neurons are not able to respond to Sonic hedgehog signals that normally induce the expression of Patched RNA, and their numbers decrease. In addition, in mouse, glial cell line–derived neurotrophic factor–related Neurturin RNA is significantly decreased. These experiments highlight the importance of parvalbumin neurons in cilium-dependent, neuroprotective signaling pathways and show that LRRK2 activation correlates with decreased Neurturin production, resulting in less neuroprotection for dopamine neurons.**

## Introduction

Parkinson's disease is characterized by the specific loss of dopamine-producing neurons that project into a region of the brain known as the dorsal striatum (Morris et al, 2024). The release of dopamine in the striatum enhances motor functions and reward signaling; in addition, dopamine plays a role in reinforcement learning (Balleine et al, 2007; Jurado-Parras et al, 2020). Imbalances in dopamine levels in the striatum are central to Parkinson's disease.

The primary cells in the striatum that respond to dopamine are known as medium spiny neurons (Balleine et al, 2007; Kreitzer & Malenka, 2008; Jurado-Parras et al, 2020). The dorsal striatum also contains multiple populations of interneurons, each comprising only a small fraction of the total cell population (Chang & Kita, 1992; Munoz-Manchado et al, 2018; Stanley et al, 2020). Among the most well-characterized striatal interneurons are the parvalbumin (PV) and cholinergic cell types (Tanimura et al, 2018; Gritton et al, 2019).

PV neurons release GABA and inhibit the activity of nearby medium spiny neurons, aiding in motor control (Koos & Tepper, 1999; Gittis et al, 2011; Tepper et al, 2018; Nahar et al, 2021). Cholinergic interneurons release acetylcholine to modulate the activity of medium spiny neurons and are also involved in motor output control. Recent studies suggest that PV neurons fine-tune the activation of medium spiny neuron networks essential for movement execution, whereas cholinergic interneurons help synchronize activity within medium spiny neuron networks, signaling the end of movement (Gritton et al, 2019).

We study an inherited form of Parkinson's disease that is caused by activating mutations in the leucine-rich repeat kinase 2 (LRRK2) (Alessi & Sammler, 2018). The LRRK2 kinase phosphorylates a subset of Rab GTPases (Steger et al, 2016, 2017), and complexes of phospho-Rab10 bound to the RILPL1 protein block the formation of primary cilia in multiple cultured cell lines (Steger et al, 2017; Dhekne et al, 2018) but only in certain cell types in the brain (Dhekne et al, 2018; Khan et al, 2021, 2024; Brahmia et al, 2024). In cell culture, ciliary blockade occurs at the earliest steps of ciliogenesis: hyperactive LRRK2 blocks the recruitment of tau tubulin kinase 2 to the mother centriole to trigger release of CP110, thus blocking cilium formation (Sobu et al, 2021).

We have shown that in the dorsal striatum in multiple LRRK2 pathway mouse models and in humans harboring *LRRK2* mutations, rare cholinergic interneurons and astrocytes lose cilia, but the much more abundant, surrounding medium spiny neurons do not (Dhekne et al, 2018; Khan et al, 2021, 2024). Loss of cilia correlates with an inability to sense and respond to Sonic hedgehog (Shh) signals (Dhekne et al, 2018) that require cilia for signal transduction (Corbit et al, 2005; Rohatgi et al, 2007; Dhekne et al, 2018). In the absence of Shh, or in the presence of pathogenic LRRK2, cholinergic interneurons decrease production of glial cell line–derived neurotrophic factor (GDNF) that normally supports dopamine neurons (Gonzalez-Reyes et al, 2012; Khan et al, 2024). In addition, Shh signaling is needed for cholinergic interneuron survival (Gonzalez-Reyes et al, 2012; Ortega-de San Luis et al, 2018).

Here, we report the consequences of *LRRK2* mutation on PV neurons that are critical regulatory contributors to the nigrostriatal circuit. We show that like cholinergic interneurons, rare PV neurons

[1]Department of Biochemistry, Stanford University School of Medicine, Stanford, CA, USA   [2]Aligning Science Across Parkinson's (ASAP) Collaborative Research Network, Bethesda, MD, USA   [3]MRC Protein Phosphorylation and Ubiquitylation Unit, University of Dundee, Scotland, UK

Correspondence: pfeffer@stanford.edu

also lose primary cilia and their ability to carry out Shh signaling; the consequence of PV neuron ciliary loss is a major reduction in GDNF-related Neurturin (NRTN) expression, and a loss of cell numbers, decreasing neuroprotection for vulnerable dopamine neurons, and contributing to Parkinson's disease.

# Results

We analyzed the primary ciliation status of PV and somatostatin interneurons in the mouse dorsal striatum of WT and *LRRK2* G2019S mice. As shown in Fig 1A and B, PV neurons were ~70% ciliated in 5-mo-old WT mice, as monitored using adenylate cyclase 3 (ACIII) antibodies to detect neuronal cilia (Dhekne et al, 2018; Sterpka & Chen, 2018). In contrast, PV neurons in the striatum of mice harboring the *LRRK2* G2019S mutation showed a roughly 30% decrease in their overall ciliation status (Fig 1A and B). In addition, the remaining cilia were ~30% shorter, which may decrease their overall signaling capacity (Fig 1D). In somatostatin interneurons detected using anti-SST28 antibodies, loss of primary cilia was slightly less pronounced at ~25% but still significant (Fig 1A and C). In this study, we focused subsequent analysis on the PV neuron class.

PV interneurons are heterogeneous: those in the dorsomedial striatum have been reported to have increased excitability compared with those in the dorsolateral striatum; they also uniquely receive glutamatergic input from the cingulate cortex (Tepper et al, 2018). Nevertheless, we detected a similar loss of cilia in both dorsolateral and dorsomedial PV neuron classes (Fig 1E).

## Ciliary loss correlates with G2019S LRRK2 expression

To investigate the basis for ciliary loss in PV neurons, we employed RNAscope fluorescence in situ hybridization to monitor the cell type–specific expression of various gene products. In these experiments, antibodies were used to detect PV neurons and their ciliation status; RNA hybridization probes were used to detect specific gene products in the identified cell types. Fig 2A shows examples of ciliated and non-ciliated cells from the WT or *LRRK2* G2019S dorsal striatum (left column) and their corresponding content of *LRRK2* mRNA (right column, white dots). Individual dots reflect amplified signals from single RNA transcripts. Quantitation of at least 100 (total) ciliated or non-ciliated PV neurons from four mice per group showed that non-ciliated WT and non-ciliated *LRRK2* G2019S PV neurons express higher levels of *LRRK2* RNA than their ciliated counterparts (Fig 2B). Thus, within the class of PV cells, higher *LRRK2* RNA correlates directly with ciliary loss. Note that G2019S LRRK2 has twice as much protein kinase activity as the WT LRRK2; thus, similar RNA levels would be expected to reflect a twofold difference in kinase activity in those cells.

## Ciliary loss correlates with loss of Sonic hedgehog signaling

Primary cilia are required to transduce the Shh signal (Corbit et al, 2005; Rohatgi et al, 2007). We thus predicted that ciliary loss should lead to the decreased expression of Shh target genes. Fig 3 shows that the expression of Patched (*Ptch1*) RNA that encodes the Shh receptor was strongly decreased in mice harboring the hyperactive G2019S LRRK2 protein (Fig 3A and B), to an extent predicted from a combination of ciliary loss and ciliary length decrease (Fig 1B and D). When the data were segregated based upon ciliation status,

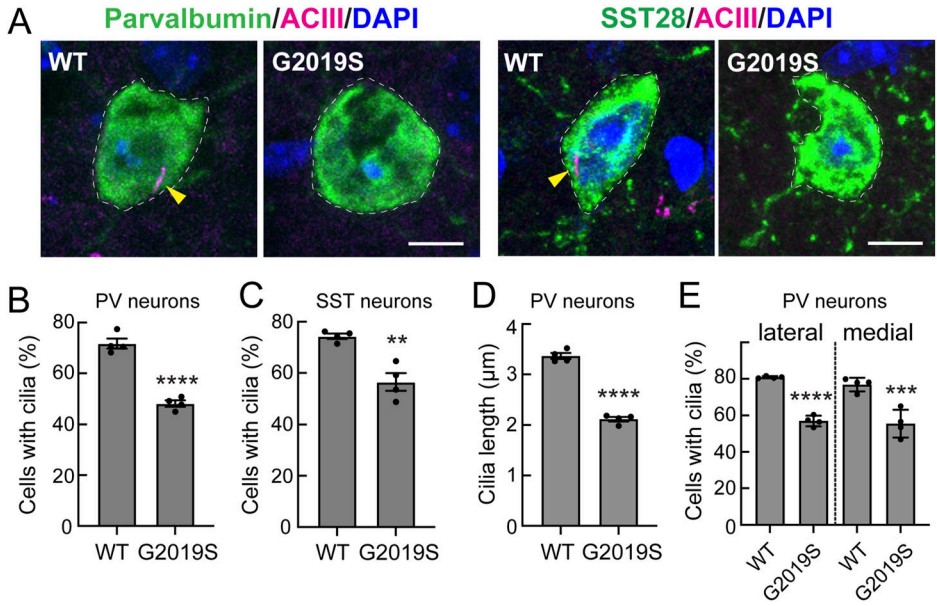

**Figure 1. G2019S LRRK2 striatal parvalbumin and somatostatin interneurons have fewer cilia.**
**(A)** Example confocal immunofluorescence micrographs of sections of the dorsal striatum from 5-mo-old WT or G2019S *LRRK2* knock-in mice; scale bar, 5 µm. (For this and all other mouse brain analysis figures, coordinates are +1.2 to −0.2 mm anterior to posterior [AP] relative to bregma.) For every cell scored, care is taken to ensure that each cilium (yellow arrowhead) is associated with a specific cell body and emanates from near the nucleus. **(A)** Left side: Parvalbumin (PV) interneuron cell bodies were detected with anti-PV antibody (green, dashed white outline); primary cilia were detected with anti-adenylate cyclase antibody (ACIII, magenta, highlighted by yellow arrowheads); nuclei were detected by DAPI (blue) staining. **(A)** Right side: Somatostatin interneurons were detected with anti-somatostatin-28 antibody (SST28, green, dashed white outline); primary cilia and nuclei were detected as for PV neurons. **(B)** Percentage of PV neurons containing a cilium. **(C)** Percentage of SST⁺ neurons containing a cilium. **(D)** Quantification of the ciliary length of PV neurons. **(B, C, D)** Significance was determined by an unpaired *t* test, (B) ****P < 0.0001; (C) **P = 0.0025; and (D) ****P < 0.0001. Values represent the mean ± SEM from individual brains, analyzing four brains per group, two to three sections per mouse, and >40 neurons per mouse. **(E)** Percentage of PV neurons containing a cilium in the dorsolateral or dorsomedial striatum for WT and G2019S mice. Significance was determined by one-way ANOVA with Tukey's test, ****P < 0.0001; ***P = 0.0001. Values represent the data (mean ± SEM) from individual brains, analyzing four brains per group, two to three sections per mouse, and >30 neurons per mouse.

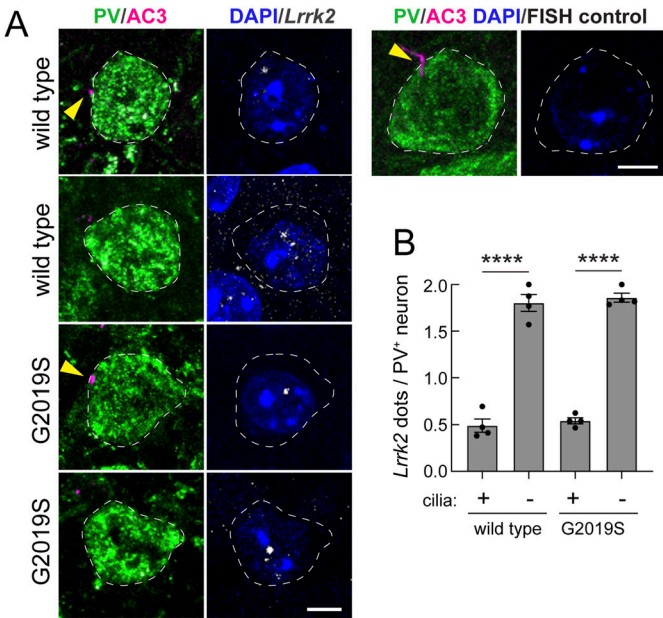

**Figure 2. Loss of primary cilia in parvalbumin interneurons correlates with higher LRRK2 expression.**
**(A)** Example confocal immunofluorescence microscopy to identify PV neurons and their cilia as in Fig 1A (left column), coupled with RNAscope in situ hybridization to detect *LRRK2* transcripts in the same cells (right column). Images of a control neuron without an RNAscope probe are shown above panel (B) at the top right. Bar, 5 μm. **(B)** Quantitation of *LRRK2* RNA dots per neuron as a function of the ciliation status for WT and G2019S *LRRK2* mice. Significance was determined by one-way ANOVA with Tukey's test, ****$P < 0.0001$. Values represent the mean ± SEM from individual brains, analyzing four brains per group, two to three sections per mouse, and >25 neurons per mouse.

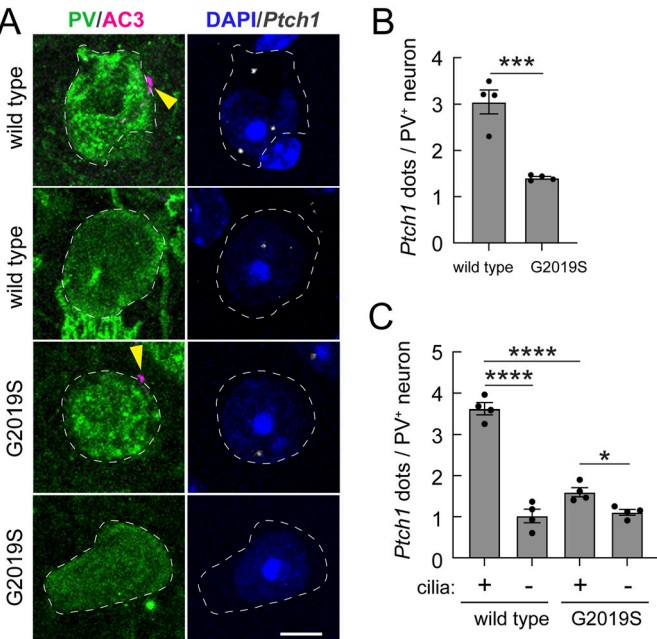

**Figure 3. Decreased cilium-dependent PTCH1 expression in striatal G2019S LRRK2 parvalbumin interneurons.**
**(A)** Example confocal immunofluorescence microscopy to identify PV neurons and their cilia as in Fig 1A (left column), coupled with RNAscope in situ hybridization to detect *Ptch1* transcripts (right column) as in Fig 2. Bar, 5 μm. **(B, C)** Quantitation of *Ptch1* RNA dots per neuron (B) or as a function of the ciliation status (C) for WT and G2019S *LRRK2* mice. **(B, C)** Significance was determined by an unpaired *t* test (B) and one-way ANOVA with Tukey's test (C). **(B)** ***$P = 0.0007$. **(C)** *$P = 0.0348$; ****$P < 0.0001$. Values represent the data (mean ± SEM) from individual brains, analyzing four brains per group, two to three sections per mouse, and >30 neurons per mouse.

ciliated WT PV neurons expressed the highest levels of *Ptch1* RNA; much less was seen in non-ciliated PV neurons (Fig 3A and C). A roughly threefold decrease in *Ptch1* expression was seen even in the ciliated G2019S *LRRK2* mutant cells, suggesting that the remaining, shorter cilia are less able to support Shh signaling. These data confirm the ciliary dependence of Shh signaling in WT striatal PV neurons and reveal a severe deficit in Shh signaling in these neurons in G2019S *LRRK2* mice.

In canonical, cilium-dependent hedgehog signaling, target genes are activated by Gli transcription factors. Overall, low Gli1 expression was detected in striatal PV neurons that were cilium-dependent in WT animals, consistent with canonical hedgehog pathways (Fig S1A–C). In the G2019S *LRRK2* striatum, we detected increased *Gli1* expression, independent of the ciliation status (Fig S1). We previously observed similar *LRRK2* mutation–linked dysregulation of *Gli1* expression in striatal cholinergic interneurons but not astrocytes (Khan et al, 2021). Further work will be needed to understand this phenomenon.

### Loss of GDNF-related Neurturin production in LRRK2 G2019S PV neurons

Neurturin (NRTN) is a member of the glial cell line–derived neurotrophic factor (GDNF) family, which plays a critical role in the survival and differentiation of dopaminergic neurons (Kotzbauer

et al, 1996). Within the striatum, NRTN is produced almost exclusively by PV neurons, with a small contribution from cholinergic neurons (Saunders et al, 2018; Dropviz.org). Because GDNF production is decreased in the absence of the Shh ligand (Gonzalez-Reyes et al, 2012) and *Gdnf* RNA is decreased in the absence of primary cilia (Khan et al, 2024), we assume that *Gdnf* is either a direct or an indirect Shh target gene. We thus explored whether PV neurons express less GDNF-related NRTN in conjunction with *LRRK2* G2019S ciliary blockade and Shh signaling dysfunction.

Fig 4A shows the results of RNAscope analysis of *Nrtn* transcripts. As predicted, we detected an almost 50% loss of *Nrtn* RNA in *LRRK2* G2019S striatal PV neurons (Fig 4B), directly analogous to the extent of Shh signaling loss (Fig 3B). When segregating the data according to the ciliation status, we noted that in WT brain, ciliated PV neurons express about three times more *Nrtn* RNA than non-ciliated PV neurons (Fig 4C). In contrast, similar to what we observed for *Ptch1* RNA expression, even ciliated PV neurons in the *LRRK2* G2019S dorsal striatum showed decreased *Nrtn* expression, consistent with a ciliary signaling defect (Fig 4C). Altogether, these data show that *LRRK2* G2019S is associated with decreased PV neuron ciliogenesis, Shh signaling, and production of the neuroprotective *Nrtn* RNA.

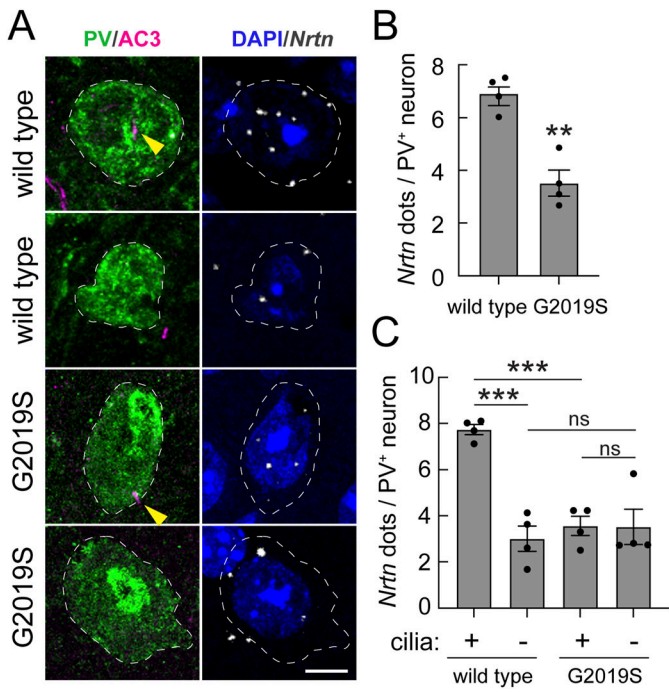

**Figure 4. Decreased cilium-dependent Neurturin expression in striatal G2019S LRRK2 parvalbumin interneurons.**

**(A)** Confocal immunofluorescence microscopy to identify PV neurons and their cilia as in Fig 1A (left column), coupled with RNAscope in situ hybridization to detect *Nrtn* transcripts (right column) as in Fig 2. Bar, 5 µm. **(B, C)** Quantitation of *Nrtn* RNA dots per neuron (B) or as a function of the ciliation status (C) for WT and G2019S *LRRK2* mice. **(B, C)** Significance was determined by an unpaired *t* test (B) and one-way ANOVA with Tukey's test (C). **(B)** **$P = 0.0011$. **(C)** ***$P = 0.0002$ between ciliated and non-ciliated WT mice; ***$P = 0.0006$ between ciliated WT and G2019S *LRRK2* mice; ns, no significance. Values represent the data (mean ± SEM) from individual brains, analyzing four brains per group, two to three sections per mouse, and >30 neurons per mouse.

## Loss of PV neurons in the LRRK2 G2019S striatum

Kottman and colleagues have shown that blocking Shh expression in dopamine neurons of the substantia nigra leads to progressive degeneration of cholinergic and PV neurons in the mouse striatum (Gonzalez-Reyes et al, 2012). A different conclusion was reached by Ortega-de San Luis et al (2018) who concluded that cholinergic but not PV neurons rely on hedgehog signaling for survival. Turcato et al (2022) concluded that blocking Shh expression in dopamine neurons does not affect striatal PV or cholinergic neuron survival; they favor models in which other Shh pools play a role. Independent of the source of Shh, we explored the consequences of ciliary loss on PV cell numbers and consequent production of neuroprotective NRTN. As shown in Fig 5A and B, the *LRRK2* G2019S dorsal striatum lost almost 40% of PV neurons across tile-scan sections of the dorsal striatum compared with WT littermates, as detected by counting PV⁺ cell bodies of four 5-mo-old mutant mice.

Cell loss was assessed more rigorously by stereological analysis of the 3-mo *LRRK2* R1441C knock-in mouse striatum. PV neurons were identified using both anti-PV and anti-cKit antibodies as an independent PV neuron marker. For this analysis, the striatum was

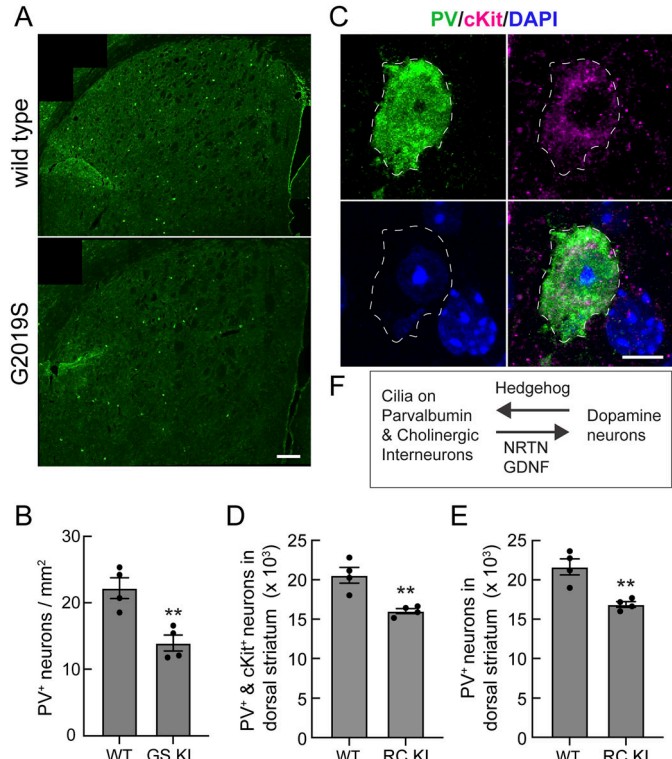

**Figure 5. Loss of parvalbumin interneurons in the G2019S and R1441C LRRK2 dorsal striatum.**

**(A)** Low-magnification tile-scan example images of total PV neurons (green) from 5-mo-old WT (upper panel) and G2019S *LRRK2* (lower panel) mice. Bar, 200 µm. **(A, B)** Quantitation of data from micrographs such as that shown in (A), detected per mm². Significance was determined by an unpaired *t* test, **$P = 0.0058$. Values represent the data (mean ± SEM) from individual brains, analyzing four brains per group and two sections per mouse, with >100 tiles scored per brain. **(C)** Example immunofluorescence micrograph of a PV⁺ and cKit⁺ neuron in the dorsal striatum of a WT mouse. PV neuron cell bodies were detected with anti-PV antibody (green, dashed white outline); the second marker for PV neurons was detected with anti-cKit antibody (magenta); nuclei were detected by DAPI (blue) staining. Bar, 5 µm. **(D, E)** Stereological estimation of double-positive ((D) PV⁺ and cKit⁺) or single-positive PV⁺-alone (E) neurons in the dorsal striatum of 3-mo-old WT and R1441C *LRRK2* mice. **(D, E)** Significance was determined by an unpaired *t* test, (D) **$P = 0.0050$; (E) **$P = 0.0045$. Values represent the data (mean ± SEM) from individual brains, analyzing four brains per group and 10 sections per mouse. **(F)** Reciprocal, cilium-dependent signaling between dopamine neurons of the substantia nigra and striatal interneurons.

cut into 60 30-µm sections that were analyzed in series, quantifying every sixth section (10 sections per mouse, 4 mice per group). The data were analyzed using Stereo Investigator software as described previously (Knowles et al, 2022) with parameters described in Table S1. As shown in Fig 5C–E, similar PV neuron loss was also detected in the R1441C *LRRK2* animals monitoring either PV⁺ and cKit⁺ neurons or PV⁺ neurons alone; the magnitude of the cell loss was less than that seen in the older *LRRK2* G2019S KI mice (Fig 5A and B). These data support the conclusion that pathogenic LRRK2 expression decreases the ability of striatal PV neurons to receive ciliary, neurotrophic Shh signals and to produce the NRTN peptide in response. This phenotype occurs concomitantly with decreased GDNF production by *LRRK2* G2019S cholinergic neurons, exacerbating the loss of neuroprotection for dopamine neurons (Fig 5F).

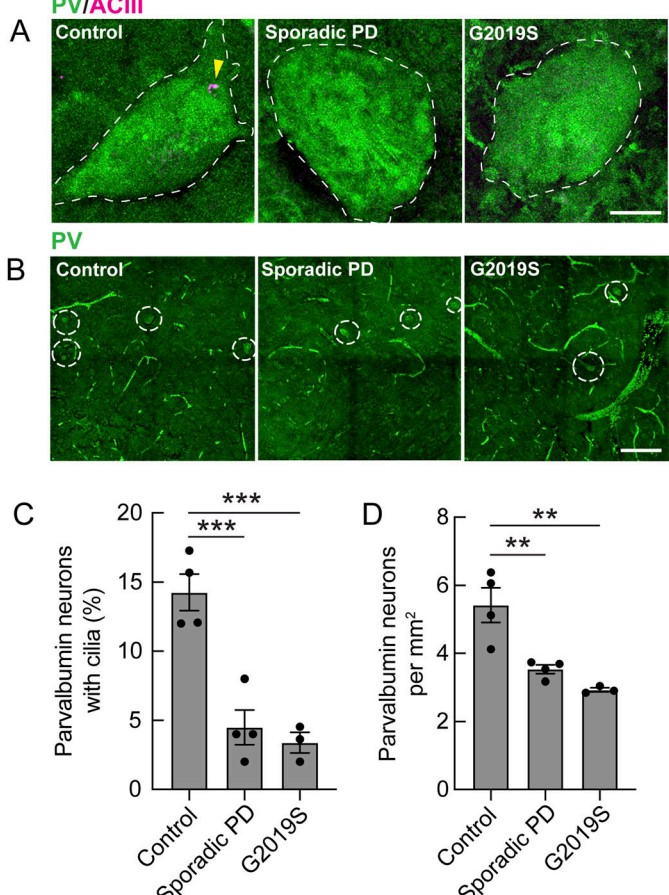

**Figure 6. Loss of primary cilia and decreased human parvalbumin interneurons in the lenticular nucleus of Parkinson's brains.**
**(A)** Confocal microscopy of the tissue labeled with anti-PV antibody (green) and anti-ACIII antibody (magenta) to identify primary cilia. Bar, 10 $\mu$m. The status of the patients is indicated in white (control, sporadic, or *LRRK2* G2019S). **(B)** Representative tile-scan images of the lenticular nucleus from postmortem brain sections from patients, identified geographically and molecularly using anti-DARPP-32 antibodies (medium spiny neuron marker); parvalbumin interneurons were identified using anti-PV antibodies (green, identified by cellular somata >30 $\mu$m). Bar, 100 $\mu$m. **(C)** Quantitation of the percentage of PV neurons containing a cilium. Error bars represent the SEM from 4 control, 4 sporadic, and 3 G2019S *LRRK2* mutation brains, with >40 PV neurons scored per brain. Statistical significance was determined using one-way ANOVA with Tukey's test. ***$P$ = 0.0007 for control versus sporadic PD and ***$P$ = 0.0005 for control versus G2019S PD. **(D)** Quantitation of human PV neurons detected per sq mm. Error bars represent the SEM from 4 control, 4 sporadic PD, and 3 G2019S PD mutation brains, with >200 tiles scored per brain. Statistical significance was determined using one-way ANOVA with Tukey's test. **$P$ = 0.0081 for control versus sporadic PD and **$P$ = 0.0025 for control versus G2019S PD.

### Loss of PV neuron cilia and cell numbers in human Parkinson's brain

We reported recently that postmortem lenticular nucleus sections from Parkinson's patients show decreased ciliation of cholinergic neurons and astrocytes but not of medium spiny neurons (Khan et al, 2024). Moreover, these changes were observed in patients harboring *LRRK2* pathway mutations, as well as patients with idiopathic Parkinson's disease. Thus, we were interested to examine

the fate of PV neurons in these brain samples. As shown in Fig 6, human lenticular nucleus PV neurons displayed a significant loss of primary cilia and we also detected an almost 50% decrease in PV cell numbers in the Parkinson's brains. Note that this analysis examined single sections per patient, as it was not possible to carry out stereological determinations because of limited sample availability.

## Discussion

Dopamine neurons innervate the striatum where they arborize extensively and secrete Shh that is sensed by PV and cholinergic interneurons. Those cells respond to Shh and provide neuroprotection to dopamine neurons by secreting NRTN and GDNF (Fig 5F). In the absence of cilia, PV and cholinergic neurons fail to produce NRTN and GDNF. They may also fail to survive in the absence of strong hedgehog signals.

We have shown here that an activating mutation in the LRRK2 kinase has profound effects on Shh signaling in PV neurons of the dorsal striatum. Ciliary loss is accompanied by a significant decrease in *Ptch1* RNA expression in PV neurons, coupled with a loss of *Nrtn* RNA. Indeed, the decrease in both *Ptch1* and *Nrtn* RNA appears to be greater than the loss of overall ciliation in this cell type, but consistent with a combination of ciliary loss and remaining ciliary shortening. Loss of Shh signaling leads to loss of PV neurons overall, consistent with prior studies showing their dependence on Shh for viability (Gonzalez-Reyes et al, 2012).

Our findings are generally consistent with the prior work of Guo et al (2017), who engineered mice to block primary cilia in PV neurons and detected a significant reduction in neuronal process complexity in the striatum, cortex, and hippocampus, without associated PV cell loss at 2 mo of age. Their data support a specific requirement for primary ciliary signaling in the morphological development of striatal interneurons, and cell death might be seen in older animals. Whether *LRRK2* mutation and associated ciliary loss also reduce neuronal process complexity will be an interesting question for future study.

A remaining puzzle is why PV and cholinergic interneurons and striatal astrocytes are vulnerable to ciliary loss, whereas the surrounding, much more abundant medium spiny neurons are not, both in mice and in humans (Dhekne et al, 2018; Brahmia et al, 2024; Khan et al, 2024). It is not because they express higher levels of *LRRK2* RNA; on the contrary, we have shown that they actually express less (Khan et al, 2024). Yet within the category of either PV interneurons (this study) or cholinergic interneuron cell types, higher *LRRK2* RNA expression correlates with decreased ciliogenesis. Further work will be needed to understand the unique vulnerability of interneuron– and astrocyte–cilia to LRRK2 kinase action. The conclusion that the ciliary phenotypes and loss of neuroprotective factor expression observed are due to LRRK2 kinase activity is strongly supported by our recent rescue of *LRRK2* mutant phenotypes upon 3-mo dietary administration of a LRRK2-specific kinase inhibitor (Jaimon et al, 2024 *Preprint*).

The present study shows that Parkinson's disease–associated *LRRK2* mutation is associated with a significant loss of *Nrtn*

expression in the mouse striatum, likely because of the loss of primary cilia in PV neurons that rely on cilia to produce NRTN in a hedgehog-responsive manner; *Gdnf* expression in cholinergic interneurons is also decreased in these animals (Khan et al, 2024). Are both neurotrophic factors equally important for dopaminergic neuron survival? Dopaminergic axons in the striatum express both the GDNF receptor, GFRA1, and the NRTN receptor, GFRA2, but with higher GFRA1 expression (Saunders et al, 2018; Dropviz.org). Even though NRTN prefers to bind GFRA2, it can still signal via GFRA1 at higher concentrations (Rossi et al, 1999). Interestingly, the NRTN GFRA2 receptor is most highly expressed in striatal PV neurons (Saunders et al, 2018; Dropviz.org), so it could contribute to a positive, autocrine feedback circuit to support PV neurons. In the future, highly sensitive mass spectrometry will be needed to determine the absolute levels of these two neuroprotective factors and their corresponding receptors in the striatum to fully understand their relative contributions to neuroprotection.

Both GDNF and NRTN have been tested as therapeutics for PD patients as part of multiple clinical trials with mixed results to date (cf. Barker et al, 2020; Chu & Kordower, 2023). A challenge for these trials has been to achieve adequate distribution of administered proteins or injected expression viruses, but alternative delivery strategies continue to be tried. These exogenous growth factors will hopefully help to sustain dopaminergic neurons. But without cilia to sense Shh, cholinergic and PV interneurons that are so important for regulating dopamine signaling will remain vulnerable and their numbers may nevertheless decrease (Gonzalez-Reyes et al, 2012). Altogether, these experiments highlight the pathogenic LRRK2-triggered loss of neuroprotection experienced by multiple neuron classes, because of a block in Shh signaling and neuroprotective factor production that has important implications for dopaminergic neuron survival in Parkinson's disease.

# Materials and Methods

Key resources are summarized in Table S3.

## Research standards for animal studies

*LRRK2*[G2019S/G2019S] and *LRRK2*[R1441C/R1441C] mice were obtained from Taconic (Constitutive KI *Lrrk2tm4.1Arte*; RRID:I MSR_TAC:13940) and form the Jackson Laboratory (B6.Cg-*Lrrk2*[tm1.1Shn]/J mouse; RRID: IMSR_JAX:009346), respectively, and kept in specific pathogen-free conditions at the University of Dundee (UK). All animal experiments were ethically reviewed and conducted in compliance with the Animals (Scientific Procedures) Act 1986 and guidelines established by the University of Dundee and the UK Home Office. Ethical approval for animal studies and breeding was obtained from the University of Dundee ethical committee, and all procedures were performed under a UK Home Office project license. The mice were group-housed in an environment with controlled ambient temperature (20–24°C) and humidity (45–55%), after a 12-h light/12-h dark cycle, with ad libitum access to SDS RM No. 3 autoclavable food and water. Genotyping of mice was performed by PCR using genomic DNA isolated from ear biopsies. Primer 1 (5′-CTGCAGGCTACTAGATGGTCAAGGT-

3′) and Primer 2 (5′-CTAGATAGGACCGAGTGTCGCAGAG-3′) were used to detect the WT and knock-in alleles (Phelan, 2023). Homozygous *LRRK2*-G2019S and *LRRK2*-R1441C mice and their littermate WT controls (5 mo old for G2019S and 3 mo old for R1441C) were produced from crossing heterozygous animals, with genotyping confirmation conducted on the day of the experiment.

## Immunohistochemical (IHC) staining for mouse brains

The mouse brain striatum (coordinates: +1.2 to −0.2 mm anterior to posterior [AP] relative to bregma) was subjected to immunostaining following a previously established protocol (Khan et al, 2020). Frozen slides were thawed at RT for 15 min and then gently washed twice with PBS for 5 min each. Antigen retrieval was achieved by incubating the slides in 10 mM sodium citrate buffer, pH 6.0, pre-heated to 95°C, for 15 min. Sections were permeabilized with 0.1% Triton X-100 in PBS at RT for 15 min, followed by blocking with 2% FBS and 1% BSA in PBS for 2 h at RT. Primary antibodies were applied overnight at 4°C, and the next day, sections were exposed to secondary antibodies at RT for 2 h. Secondary antibodies used were donkey highly cross-absorbed H + L antibodies conjugated to Alexa 488 or CF488, Alexa 568, or Alexa 647, diluted at 1:2000. Nuclei were counterstained with 0.1 μg/ml DAPI (Sigma-Aldrich). Finally, stained tissues were mounted with Fluoromount-G and covered with a glass coverslip. All antibody dilutions for tissue staining contained 1% DMSO to facilitate antibody penetration.

## Fluorescence in situ hybridization

RNAscope fluorescence in situ hybridization for the mouse brain striatum (coordinates: +1.2 to −0.2 mm anterior to posterior [AP] relative to bregma) was carried out as described previously (Khan et al, 2021). RNAscope Multiplex Fluorescent Detection Kit v2 (Advanced Cell Diagnostics) was used following the manufacturer's instructions, employing RNAscope 3-plex Negative Control Probe (#320871) or Mm-Lrrk2 (#421551), Mm-Gli1 (#311001), Mm-Ptch1-C2 (#402811-C2), and Mm-Nrtn-C2 (#441501-C2). The Mm-Lrrk2, Mm-Ptch1-C2, and Mm-Nrtn-C2 probes were diluted 1:20, 1:5, and 1:3, respectively, in dilution buffer consisting of 6x saline/sodium citrate buffer (SSC), 0.2% lithium dodecyl sulfate, and 20% Calbiochem OmniPur Formamide. Fluorescent visualization of hybridized probes was achieved using Opal 690 (Akoya Biosciences). Subsequently, brain slices were subjected to blocking with 1% BSA and 2% FBS in TBS (Tris-buffered saline) with 0.1% Triton X-100 for 30 min. They were then exposed to primary antibodies overnight at 4°C in TBS supplemented with 1% BSA and 1% DMSO. Secondary antibody treatment followed, diluted in TBS with 1% BSA and 1% DMSO containing 0.1 μg/ml DAPI (Sigma-Aldrich) for 2 h at RT. Finally, sections were mounted with Fluoromount-G and covered with glass coverslips.

## Unbiased stereological microscopy and the optical fractionator method

Coronal mouse brain sections (30 μm) containing the striatum were processed for immunostaining and cell counting using unbiased stereology with Stereo Investigator, following a previously established protocol (Cenera, 2024) and the published work of

Knowles et al (2022). Parvalbumin neurons, labeled with anti-parvalbumin and anti-cKit antibodies, were visualized using an MBF Zeiss Axiocam light microscope. Cell counts were obtained via unbiased stereology with Stereo Investigator software (MBF Bioscience, version 2024). Regions of interest, which included the dorsal striatum (caudate putamen) from +1.2 to −0.6 mm anterior to posterior (AP) relative to bregma in mice, were traced at 2.5X magnification based on dopamine receptor D1 staining. The nucleus accumbens was excluded from all analyses. Images were captured at 40X magnification from every sixth section across this region, including 10 sections per animal. No differences were observed in the number of sections used between experimental groups. Stereological parameters, as shown in Table S1, were based on pilot studies and literature (Gonzalez-Reyes et al, 2012; Filice et al, 2016). The total number of neurons was estimated using the optical disector formula (see Oorschot [1996]). The Gunderson m = 1 coefficient of error (CE) was calculated following the guidelines of Gundersen and Jensen, with values below 0.10 considered acceptable for biological samples (Gundersen & Jensen, 1987; Slomianka & West, 2005; West, 2013). Exposure time was kept consistent across all samples imaged within each experiment. Unless otherwise noted, a disector height of 30 $\mu$m and guard zones of 3 $\mu$m were applied in all studies.

### Immunohistochemical (IHC) staining for human brains

Human brain sections were obtained from the Banner Sun Health Research Institute Brain and Body Donation Program of Sun City, Arizona, from elderly volunteers; written informed consent was obtained for their use. Immunohistochemistry of the human brain lenticular nucleus was conducted as previously described (Khan et al, 2023). Briefly, free-floating tissues were subjected to antigen retrieval by incubating them in 10 mM sodium citrate buffer (pH 6.0), preheated to 95°C, for 15 min at 95°C. After this, tissues were permeabilized with 0.1% Triton X-100 in PBS at RT for 1 h. They were then blocked in a solution of 2% FBS and 1% BSA in PBS for 2 h at RT, after which they were incubated overnight at 4°C with primary antibodies. The next day, tissues were incubated with secondary antibodies for 2 h at RT. Secondary antibodies were donkey highly cross-absorbed H+L conjugated to Alexa 488, Alexa 568, or Alexa 647, used at a 1:2,000 dilution. Nuclei were counterstained with 0.1 $\mu$g/ml DAPI (Sigma-Aldrich). To minimize autofluorescence, tissues were incubated with freshly prepared 0.1% Sudan Black B in 70% ethanol for 20 min. Stained tissues were then transferred to slides, mounted with Fluoromount-G, and covered with a glass coverslip. All antibody dilutions contained 1% DMSO to enhance antibody penetration. Images were captured using a Zeiss LSM 900 confocal microscope with either a 63×/1.4 oil immersion objective or a 20×/0.8 objective. Image visualization and analysis were conducted using Fiji software. Table S2 summarizes the areas imaged for each sample.

### Microscope image acquisition

All images were obtained using a Zeiss LSM 900 confocal microscope (Axio Observer Z1/7) coupled with an Axiocam 705 camera and an immersion objective (Plan-Apochromat 63x/1.4

Oil DIC M27) or objectives (Plan-Apochromat 20x/0.8 M27 and EC Plan-Neofluar 5x/0.16 M27). The images were acquired using ZEN 3.4 (blue edition) software, and visualizations and analyses were performed using Fiji.

In addition to the above-mentioned methods, all other statistical analysis was carried out using GraphPad Prism version 9.3.1 for Macintosh (GraphPad Software, Boston, Massachusetts, USA; www.graphpad.com).

## Data Availability

All primary data are available at Yu-En et al (2024); https://doi.org/10.5061/dryad.z08kprrpx and Lin et al (2024); https://doi.org/10.5281/zenodo.11510059.

## Supplementary Information

## Acknowledgements

This study was funded by the joint efforts of the Michael J Fox Foundation for Parkinson's Research (MJFF) and the Aligning Science Across Parkinson's (ASAP) initiative. MJFF administers the grant (ASAP-000463) on behalf of ASAP and itself. For the purpose of open access, the authors have applied a CC-BY public copyright license to the Author Accepted Manuscript version arising from this submission. We are grateful to the Banner Sun Health Research Institute Brain and Body Donation Program of Sun City, Arizona, for the provision of human biological materials. The Brain and Body Donation Program has been supported by the National Institute of Neurological Disorders and Stroke (U24 NS072026 National Brain and Tissue Resource for Parkinson's Disease and Related Disorders), the National Institute on Aging (P30AG19610 and P30AG072980, Arizona Alzheimer's Disease Center), the Arizona Department of Health Services (Contract 211002, Arizona Alzheimer's Research Center), the Arizona Biomedical Research Commission (Contracts 4001, 0011, 05-901, and 1001 to the Arizona Parkinson's Disease Consortium), and the Michael J Fox Foundation for Parkinson's Research. We also thank Drs. Haojun Xu and Michelle Monje at Stanford University for their assistance and for providing access to the stereology microscopy.

### Author Contributions

Y-E Lin: conceptualization, data curation, formal analysis, investigation, visualization, methodology, and writing—review and editing.
E Jaimon: conceptualization, data curation, formal analysis, investigation, visualization, methodology, and writing—review and editing.
F Tonelli: resources and project administration.
SR Pfeffer: conceptualization, data curation, formal analysis, supervision, funding acquisition, visualization, project administration, and writing—original draft.

### Conflict of Interest Statement

The authors declare that they have no conflict of interest.

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
