## [Reviewer comments · Life Science Alliance]

Life Science Alliance

Pathogenic LRRK2 mutations cause loss of primary cilia and NRTN in striatal Parvalbumin interneurons

Yu-En Lin, Ebsy Jaimon, Francesca Tonelli, and Suzanne Pfeffer

DOI: <https://doi.org/10.26508/lsa.202402922>

Corresponding author(s): Suzanne Pfeffer, Stanford University

Review Timeline:

Submission Date:	2024-07-02
Editorial Decision:	2024-08-05
Revision Received:	2024-10-19
Editorial Decision:	2024-10-21
Revision Received:	2024-10-25
Accepted:	2024-10-28

Transaction Report:

August 5, 2024

Re: Life Science Alliance manuscript #LSA-2024-02922-T

Dr. Suzanne R. Pfeffer
Stanford University
Dept. of Biochemistry
School of Medicine
Stanford, 279 Campus Drive B400 94305-5307

Dear Dr. Pfeffer,

Thank you for submitting your manuscript entitled "Pathogenic LRRK2 mutations cause loss of primary cilia and NRTN in striatal Parvalbumin interneurons" to Life Science Alliance. The manuscript was assessed by expert reviewers, whose comments are appended to this letter. We invite you to submit a revised manuscript addressing the Reviewer comments.

Thank you for this interesting contribution to Life Science Alliance. We are looking forward to receiving your revised manuscript.

Sincerely,

B. MANUSCRIPT ORGANIZATION AND FORMATTING:

Reviewer #1 (Comments to the Authors (Required)):

The manuscript by Lin and colleagues shows that parvalbumin (PV) interneurons in the dorsal striatum lose primary cilia in G2019S LRRK2 knockin mice. As a consequence, these neurons are unable to respond to sonic hedgehog signals as measured by decreased mRNA levels for patched1. The authors further show a decrease in neurturin mRNA levels which may impact dopaminergic health, and a decrease in PV interneuron cell numbers. These data are novel and interesting and warrant publication in Life Science Alliance. However, I have several major concerns which need addressing:

1. Figure 2: the authors show that cells which lack cilia express more LRRK2 as measured by RNAscope. This is observed in PV neurons from both wildtype and G2019S LRRK2-KI mice. Does this mean that wt LRRK2 also blocks ciliogenesis when overexpressed/expressed to a higher extent? And does higher wt LRRK2 expression in non-ciliated cells then also cause death of these neurons? Has an effect of wt LRRK2 overexpression on ciliogenesis been shown in vitro or in vivo? Please explain and discuss. On page 4, the authors mention that quantification was performed from at least 100 ciliated or non-ciliated PV neurons from four mice. Does this mean 100 cells per mouse, or 100 cells from 4 mice (25 cells per mouse)? Please clarify.
2. The authors show an overall decrease in Ptch1 dots/PV neuron in G2019S LRRK2-KI mice, indicative of defective Shh signaling (Figure 3). What about Gli1 as a measure of Shh signaling, which the authors have used previously? Please at least discuss why Ptch1 rather than Gli1 was chosen for RNAscope analysis.
3. Figure 4 shows a loss of neurturin (NRTN) as measured by RNAscope in PV neurons from G2019S LRRK2-KI mice. Has it been previously demonstrated that NRTN expression is induced upon Shh signaling in those neurons? Given previous reports that PV neurons are the main producers of GDNF in the striatum (Hidalgo-Figueroa et al., J. Neurosci., 2012), the authors should also measure GDNF in the PV neurons. Do dopaminergic neuronal axons in the striatum contain GFRalpha2 (NRTN receptor) and/or GFRalpha1 (GDNF receptor)? Is NRTN required for the survival of the PV neurons, or is its role limited to protecting dopaminergic neurons? A better description of what is known about NRTN and GDNF targets in the striatum is warranted.
4. The authors show a decrease in the number of PV interneurons in dorsal striatum from 5-months old G2019S LRRK2-KI mice. In which area of the dorsal striatum was this determined (Bregma coordinates)? Co-staining with another marker to assure cell death, rather than a decrease in parvalbumin expression, is warranted. Analysis over two sections per animal, given the inherent variability of cell count/density determinations, is underpowered. As a more accurate measure, the authors may want to plot total cell number as determined across different sections (e.g. Gonzalez-Rodriguez, P. et al., Nature 599, 650-656 (2021)).
5. Is there evidence for PV neuronal cell death in dorsal striatum in human PD? Ciliary loss in PV neurons may have a negative impact upon dopaminergic cell health, but this may occur irrespective of whether PV neurons die or not.
6. The authors cite a study showing that Shh from dopaminergic neurons triggers neurotrophic factor production in striatal cholinergic interneurons and PV interneurons to support dopaminergic neuron survival (Gonzalez-Reyes et al., 2012). However, these findings have been challenged by two subsequent reports (Ortega-de San Luis et al., Aging Cell, 2018; Turcato et al., iScience, 2022). The authors should at least discuss these discrepancies and highlight potential other mechanisms that could explain their data.
7. Methods section: In many cases, rudimentary details of methods are given in the figure legends rather than in the methods section. In the methods section, there is no detailed description of how animals were perfused, cryosectioning details, how % ciliation and ciliary lengths were determined, or determination of cell counts/density. In addition, a careful annotation of Bregma coordinates is required in all cases to indicate where in the striatum the cilia and cell count determinations were performed.

Reviewer #2 (Comments to the Authors (Required)):

The study presented in the manuscript by Lin et al. analyzed how LRRK2 affects striatal parvalbumin positive interneurons. To

do this, ciliation rates, cilia length, expression levels of LRRK2 and markers of sonic hedgehog signaling, as well as parvalbumin neuron numbers were measured in WT and G2019S transgenic mouse brains. The authors propose that LRRK2 affects ciliation, Shh signaling and neurturin expression in these neurons. In addition, the data points to a correlation between LRRK2 expression and ciliation and to a loss of parvalbumin neurons in G2019S transgenic mice. This study is important as it specifically examines how LRRK2 affects ciliation in an important interneuron subtype of the striatum and reveals a mechanism of how these phenomena may lead to dopaminergic neuron loss in Parkinson's disease. The study is well organized and experimental evidence is well presented, however some additions or adjustments to experimental procedures would be welcome to reinforce some of the observations made.

Concerning the measures of LRRK2 expression, these are mRNA measures. Given the conclusions that LRRK2 expression is linked to alterations in ciliation, it would be beneficial to add a measure of LRRK2 protein expression, replacing the RNAscope with an anti-LRRK2 immunohistochemical detection.

Concerning the measures of parvalbumin neuron number, where the authors report a decreased number of striatal parvalbumin positive neurons in the striatum expressed as a number per tissue surface, it is absolutely essential to proceed with a stereological approach to obtain parvalbumin neuron numbers per striatal volume. Also, details should be given of which region of the striatum has been analyzed along its rostro-caudal axis (the image given appear to be in a caudal section of the striatum) as well as how the striatum was delineated. For instance, is the analysis inclusive of the nucleus accumbens or is this exclusively the caudate-putamen?

Minor comments

- Please provide references for the sentence in the introduction 'Recent studies suggest that PV neurons fine-tune the activation of medium spiny neuron networks essential for movement execution, whereas cholinergic interneurons help synchronize activity within medium spiny neuron networks, signaling the end of movement.'
- Please include a reference or references and include a some explanatory text concerning the choice of ACIII as a marker of neuronal primary cilia.
- This reviewer would suggest modifying the sentence stating that dopaminergic neurons 'infiltrate' the striatum, given the connotation of the word infiltrate that suggest gaining access surreptitiously. Perhaps the term 'innervate' may be an alternative?

Reviewer #3 (Comments to the Authors (Required)):

As previously shown by the authors, the Lrrk2 G2019S-induced reduction of primary cilia was only observed in a limited number of cells in the striatum and has previously been observed in acetylcholine neurons and astrocytes. In this manuscript, Pfeffer and colleagues showed that in striatal parvalbumin neurons of Lrrk2 G2019S knockin (KI) mice, the proportion with primary cilia was also reduced and their length was shorter (Fig.1,2). PV neurons with primary cilia had higher expression for the Shh receptor Ptch1 (Fig.3). There was also greater expression of neurturin (Fig.4). In addition, PV neurons were reduced by 40% in the striatum of G2019S KI mice (Fig.5). These results support the authors' hypothesis that the LRRK2 G2019S mutation reduces the proportion of certain cells in the striatum with primary cilia, thereby suppressing downstream GDNF and NRTN expression and disrupting DA neurons due to defects in Shh signaling. The hypothesis derived from the correlation between neuronal subtypes and the presence or absence of primary cilia and gene expression profiles, including Lrrk2, is attractive, but causal relationship between the observed phenomena, such as whether Shh signaling is incomplete in PV neurons without primary cilia or whether NRTN expression is low in cells with low Ptch1 expression, have not been demonstrated.

Major:

1. The authors compare mice of different genotypes, but are they from the same strain? It should be clarified whether they are comparing WT and homozygous G2019S KI produced by crossing heterozygous G2019S KI, or just hybrid WT and homozygous KI bred independently in the same institution. Experiments are also needed to compare the number of PV neurons with primary cilia by administering MLI-2 to homozygous G2019S KI.
2. The quantitative nature of RNAscope is unclear. Gene expression levels are quantified by the number of dots in the RNA hybridization, and it is unclear whether this correlates with actual expression levels. If there is one spot, does that mean there is one mRNA?
3. The expression of factors regulated downstream of Shh signaling in PV neurons should also be quantified and related to the presence of primary cilia. In Dhekne et al., eLife, 2018, the authors showed, albeit in a cultured cell system, that Gli1 expression was decreased in LRRK2 G2019S het iPS cells.
4. To clarify the causal relationship between Lrrk2 and Ptch1 and Nrtin expression changes, Lrrk2, Ptch1 and Nrtin should be RNAscope-stained in the same sections to clarify the association in their expression levels.

Minor:

1. Fig. 1: There is no explanation of why the authors check for somatostatin interneurons as well.
2. p.4 The description of Fig. 2B says "higher levels of LRRK2 enzyme", but it should read "higher numbers of Lrrk2 dots", since the authors did not confirm the change in protein expression or the enzymatic activity.

We thank the reviewers for their constructive comments and have addressed essentially all of the comments with significant additional data. The changes have truly improved the manuscript. Thank you!

Reviewer #1 The manuscript by Lin and colleagues shows that parvalbumin (PV) interneurons in the dorsal striatum lose primary cilia in G2019S LRRK2 knockin mice. As a consequence, these neurons are unable to respond to sonic hedgehog signals as measured by decreased mRNA levels for patched1. The authors further show a decrease in neurturin mRNA levels which may impact dopaminergic health, and a decrease in PV interneuron cell numbers. These data are novel and interesting and warrant publication in Life Science Alliance. **THANK YOU** However, I have several major concerns which need addressing:

1. Figure 2: the authors show that cells which lack cilia express more LRRK2 as measured by RNAscope. This is observed in PV neurons from both wildtype and G2019S LRRK2-KI mice. Does this mean that wt LRRK2 also blocks ciliogenesis when overexpressed/expressed to a higher extent? *Within cholinergic and PV interneurons in brain, whether they are wild type or mutant, non-ciliated cells express less LRRK2. LRRK2 G2019S has twice the kinase activity as wild type LRRK2 so equal expression of the two protein forms has very different consequences. We conclude that LRRK2 expression correlates with cilia loss and mutant LRRK2 expression correlates with more cilia loss.*

And does higher wt LRRK2 expression in non-ciliated cells then also cause death of these neurons? Not that we know of. Lack of Hedgehog signaling was shown by Kottmann to lead to CIN and PV cell death and dopaminergic neuron cell death; we detect cell death under conditions where cilia (and thus Hedgehog signaling) are decreased. While we see a decrease in PV cell number in the mutant animals, we have not proven that this was caused by cilia loss. We have modified the text throughout to avoid over-interpretation of the findings.

*We added this information to the text: Guo and colleagues PMID: 28787594 generated *Parvcre; Arl13b^{lox/lox}; Ai9 (Parv-Arl13b; Ai9)* mice to inactivate primary cilia signaling in PV neurons from postnatal week 2, after completion of interneuron generation and localization. Evident by P60 was a significant reduction in PV interneuron process complexity in the striatum, cortex, and hippocampus, without associated PV cell loss. These data support a specific requirement for primary ciliary signaling in the morphological development of striatal interneurons; cell death might be seen in older animals.*

Has an effect of wt LRRK2 overexpression on ciliogenesis been shown in vitro or in vivo? Please explain and discuss. We have not carefully overexpressed wt LRRK2 in cell culture and scored cilia, but we have looked at ciliation in PPM1H knockout cells—these are missing the phosphoRab-specific phosphatase. In these cells, cilia are lost due to failure to reverse the action of their wild type LRRK2 (Berndsen et al., PMID: 31663853). In vivo, mutant LRRK2 BAC transgene overexpression (~6X) and LRRK2 knockin mutation show same cilia phenotype for ChAT neurons and astrocytes (PMC8550758) but within a given cell type, more LRRK2 RNA correlates with less ciliation as discussed above (PMC11317616).

On page 4, the authors mention that quantification was performed from at least 100 ciliated or non-ciliated PV neurons from four mice. Does this mean 100 cells per mouse, or 100 cells from 4 mice (25 cells per mouse)? Please clarify. The quantification was performed from 25 PV neurons in the dorsal striatum per mouse—we clarified the text.

2. The authors show an overall decrease in Ptch1 dots/PV neuron in G2019S LRRK2-KI mice, indicative of defective Shh signaling (Figure 3). What about Gli1 as a measure of Shh signaling, which the authors have used previously? Please at least discuss why Ptch1 rather than Gli1 was chosen for RNAscope

analysis. We previously used Gli1 as a readout of Shh signaling. However, as now shown in the new **Supplementary Figure 1**, there is an overshooting of overall Gli1 transcript in PV neurons of LRRK2-G2019S mice. Also, dysregulated Gli1 upregulation found in G2019S mice did not track with the ciliation status in PV neurons that is seen in wild type animals. These phenomena mirror that of our previous publication analyzing Gli1 regulation in cholinergic neurons (Khan et al., eLife, 2021). We now show both Gli1 and PTCH1 but PTCH1 is a less complicated readout in this system and discuss this.

3. Figure 4 shows a loss of neurturin (NRTN) as measured by RNAscope in PV neurons from G2019S LRRK2-KI mice. Has it been previously demonstrated that NRTN expression is induced upon Shh signaling in those neurons? No, it has not been shown to our knowledge for striatal PV neurons, but it is not very surprising since NRTN is a GDNF relative that is known to be induced by Hh signaling—we discuss this.

Given previous reports that PV neurons are the main producers of GDNF in the striatum (Hidalgo-Figueroa et al., J. Neurosci., 2012), the authors should also measure GDNF in the PV neurons. Do dopaminergic neuronal axons in the striatum contain GFRalpha2 (NRTN receptor) and/or GFRalpha1 (GDNF receptor)?

According to more recent <http://dropviz.org/> transcriptomics, striatal ChAT neurons produce 3X more GDNF RNA than PV neurons and PV neurons produce 4X more NRTN RNA than ChAT neurons. Thus, we do not feel that it adds to our story to monitor GDNF in PV neurons. Dopaminergic axons in the striatum contain both receptors, with higher GFRalpha1 expression levels (Dropviz). Despite the fact that NRTN prefers its receptor, GFRA2, NRTN can still signal via GDNF receptor, GFRA1, at high concentrations (Rossi et al., Neuron, 1999). We have added this to the text.

Is NRTN required for the survival of the PV neurons, or is its role limited to protecting dopaminergic neurons? A better description of what is known about NRTN and GDNF targets in the striatum is warranted. We have expanded the discussion. Interestingly, the NRTN GFRA2 receptor is most highly expressed on striatal PV neurons (Dropviz. org) so it could contribute to positive, autocrine circuit.

4. The authors show a decrease in the number of PV interneurons in dorsal striatum from 5-months old G2019S LRRK2-KI mice. In which area of the dorsal striatum was this determined (Bregma coordinates)? Co-staining with another marker to assure cell death, rather than a decrease in parvalbumin expression, is warranted. Analysis over two sections per animal, given the inherent variability of cell count/density determinations, is underpowered. As a more accurate measure, the authors may want to plot total cell number as determined across different sections (e.g. Gonzalez-Rodriguez, P. et al., Nature 599, 650-656 (2021)).

We thank the reviewer for holding us to the highest standard. We analyzed an entirely new set of brains and counted the total number of PV and cKit double positive neurons (or single positive PV+ neurons) in the dorsal striatum (caudate putamen, coordinates: +1.20 mm (anterior to bregma) to -0.60 mm (posterior to bregma), with 30- μ m thickness per section, 6 section interval, 10 sections per mouse); this was quantified and plotted in **new Figure 5**, based on stereological microscopy as suggested by reviewers. The Stereological sampling parameters are shown in **Supplementary Table 1**.

5. Is there evidence for PV neuronal cell death in dorsal striatum in human PD? Ciliary loss in PV neurons may have a negative impact upon dopaminergic cell health, but this may occur irrespective of whether PV neurons die or not.

Agreed. As shown in **newly added Figure 6 (and also supplementary Table 2)**, we observe PV neuron cilia loss and cell number decrease in the lenticular nucleus (human striatum) in both sporadic and G2019S PD patients.

We agree that PV neurons need not die to fail to provide neuroprotection; we just noted that they die.

6. The authors cite a study showing that Shh from dopaminergic neurons triggers neurotrophic factor production in striatal cholinergic interneurons and PV interneurons to support dopaminergic neuron survival (Gonzalez-Reyes et al., 2012). However, these findings have been challenged by two subsequent reports (Ortega-de San Luis et al., Aging Cell, 2018; Turcato et al., iScience, 2022). The authors should at least discuss these discrepancies and highlight potential other mechanisms that could explain their data. Thank you for this comment. We have added these references and discussed them as requested. The controversy is related to the source of Shh and vulnerability of PV neurons.

7. Methods section: In many cases, rudimentary details of methods are given in the figure legends rather than in the methods section. In the methods section, there is no detailed description of how animals were perfused, cryosectioning details, how % ciliation and ciliary lengths were determined, or determination of cell counts/density. In addition, a careful annotation of Bregma coordinates is required in all cases to indicate where in the striatum the cilia and cell count determinations were performed.

We have provided links to highly detailed Protocols.io written for every method we employed. In addition, we clarified that all data collected from the dorsal striatum (caudate putamen) covers coordinates from +1.20 mm anterior to bregma to -0.20 mm posterior to bregma. For stereology, the dorsal striatum (caudate putamen) covers coordinates from +1.20 mm anterior to bregma to -0.60 mm posterior to bregma.

Reviewer #2 (Comments to the Authors (Required)):

The study presented in the manuscript by Lin et al. analyzed how LRRK2 affects striatal parvalbumin positive interneurons. To do this, ciliation rates, cilia length, expression levels of LRRK2 and markers of sonic hedgehog signaling, as well as parvalbumin neuron numbers were measured in WT and G2019S transgenic mouse brains. The authors propose that LRRK2 affects ciliation, Shh signaling and neurturin expression in these neurons. In addition, the data points to a correlation between LRRK2 expression and ciliation and to a loss of parvalbumin neurons in G2019S transgenic mice. This study is important as it specifically examines how LRRK2 affects ciliation in an important interneuron subtype of the striatum and reveals a mechanism of how these phenomena may lead to dopaminergic neuron loss in Parkinson's disease. The study is well organized and experimental evidence is well presented, however some additions or adjustments to experimental procedures would be welcome to reinforce some of the observations made. **THANK YOU**

Concerning the measures of LRRK2 expression, these are mRNA measures. Given the conclusions that LRRK2 expression is linked to alterations in ciliation, it would be beneficial to add a measure of LRRK2 protein expression, replacing the RNAscope with an anti-LRRK2 immunohistochemical detection. Despite detection of LRRK2 RNA, it is extremely poorly expressed in these cells and current methods to monitor protein reliably at these low levels do not work. Our data hold for rare cell subtypes and the RNAscope provides single cell level analysis that is highly revealing and very distinct among neuron types. Importantly, increased LRRK2 RNA levels correlate with loss of cilia that is reversed by LRRK2 inhibitor and thus due to LRRK2 activity—we clarify this in the text

bioRxiv 2024.07.31.606089; doi: <https://doi.org/10.1101/2024.07.31.606089>

Concerning the measures of parvalbumin neuron number, where the authors report a decreased number of striatal parvalbumin positive neurons in the striatum expressed as a number per tissue surface, it is absolutely essential to proceed with a stereological approach to obtain parvalbumin neuron numbers per striatal volume. **WE HAVE DONE THIS NOW**. Also, details should be given of which region of the

striatum has been analyzed along its rostral-caudal axis (the image given appear to be in a caudal section of the striatum) as well as how the striatum was delineated. For instance, is the analysis inclusive of the nucleus accumbens or is this exclusively the caudate-putamen?

The total number of PV and cKit double positive neurons in the dorsal striatum (caudate putamen, coordinates: +1.20 mm (anterior to bregma) to -0.60 mm (posterior to bregma), with 30- μ m thickness per section, 6 section interval, 10 sections per mouse) was quantified and plotted in **new Figure 5**, based on the stereological microscopy as suggested by reviewers. The Stereological sampling parameters are shown in **Table 1**. The dorsal striatum (caudate putamen) was delineated by the D1DR staining, and the nucleus accumbens were excluded from all of our analyses.

Minor comments

- Please provide references for the sentence in the introduction 'Recent studies suggest that PV neurons fine-tune the activation of medium spiny neuron networks essential for movement execution, whereas cholinergic interneurons help synchronize activity within medium spiny neuron networks, signaling the end of movement.'

We have added: Gritton, et al. (2019).

- Please include a reference or references and include a some explanatory text concerning the choice of ACIII as a marker of neuronal primary cilia.

This is a commonly used marker; we have added: Sterpka A, Chen X. Neuronal and astrocytic primary cilia in the mature brain. Pharmacol Res. 2018 Nov;137:114-121. PMID: 30291873

- This reviewer would suggest modifying the sentence stating that dopaminergic neurons 'infiltrate' the striatum, given the connotation of the word infiltrate that suggest gaining access surreptitiously. Perhaps the term 'innervate' may be an alternative? We have changed the text as requested.

Reviewer #3 (Comments to the Authors (Required)): As previously shown by the authors, the Lrrk2 G2019S-induced reduction of primary cilia was only observed in a limited number of cells in the striatum and has previously been observed in acetylcholine neurons and astrocytes. In this manuscript, Pfeiffer and colleagues showed that in striatal parvalbumin neurons of Lrrk2 G2019S knockin (KI) mice, the proportion with primary cilia was also reduced and their length was shorter (Fig.1,2). PV neurons with primary cilia had higher expression of the Shh receptor Ptch1 (Fig.3). There was also greater expression of neurturin (Fig.4). In addition, PV neurons were reduced by 40% in the striatum of G2019S KI mice (Fig.5). These results support the authors' hypothesis that the LRRK2 G2019S mutation reduces the proportion of certain cells in the striatum with primary cilia, thereby suppressing downstream GDNF and NRTN expression and disrupting DA neurons due to defects in Shh signaling. The hypothesis derived from the correlation between neuronal subtypes and the presence or absence of primary cilia and gene expression profiles, including Lrrk2, is attractive, but causal relationship between the observed phenomena, such as whether Shh signaling is incomplete in PV neurons without primary cilia or whether NRTN expression is low in cells with low Ptch1 expression, have not been demonstrated.

Major:

1. The authors compare mice of different genotypes, but are they from the same strain? It should be clarified whether they are comparing WT and homozygous G2019S KI produced by crossing heterozygous G2019S KI, or just hybrid WT and homozygous KI bred independently in the same institution. The mice are littermates produced from crossing heterozygous animals. We have clarified the text.

Experiments are also needed to compare the number of PV neurons with primary cilia by administering MLI-2 to homozygous G2019S KI. Administration of MLI-2 is a major and expensive experiment and presented in another preprint—the cilia do come back. MLI-2 feeding for 2 weeks did not rescue cilia (Dhekne et al., 2018) but feeding for 3 months does ([www.biorxiv.org/content/10.1101/2024.07.31.606089v1.full.pdf](http://www.biorxiv.org/content/10.1101/2024.07.31.606089v1.full.pdf))

2. The quantitative nature of RNAscope is unclear. Gene expression levels are quantified by the number of dots in the RNA hybridization, and it is unclear whether this correlates with actual expression levels. If there is one spot, does that mean there is one mRNA? RNA quantification is challenging but yes, one spot correlates with one RNA molecule. The method uses amplification to detect RNA and we titrate the probe so that we do not saturate the signal. Our labeling intensities match cell type specific, relative expression obtained in single nucleus RNA seq experiments such as Khan et al., 2024 (PMID: 39088390)

3. The expression of factors regulated downstream of Shh signaling in PV neurons should also be quantified and related to the presence of primary cilia. In Dhekne et al., eLife, 2018, the authors showed, albeit in a cultured cell system, that Gli1 expression was decreased in LRRK2 G2019S het iPS cells. Here we have used PTCH1 as a downstream marker rather than Gli1 as Gli1 responses can be either activating or inhibitory and thus more complex. We have now also included Gli1 data as a new Supplementary Figure 1. In this case, in wild type cells, Gli1 RNA is higher in ciliated cells compared with non-ciliated cells; in the mutant, Gli1 is induced even in non-ciliated cells. Such an overshoot was also seen in our previous analysis of cholinergic interneurons in R1441C LRRK2 mouse striatum.

4. To clarify the causal relationship between Lrrk2 and Ptch1 and Nrtn expression changes, Lrrk2, Ptch1 and Nrtn should be RNAscope-stained in the same sections to clarify the association in their expression levels.

This experiment is technically not possible because we would need 5 channels to score these plus the cell type specific marker and either the cilia marker or DAPI. Our data show: LRRK2 expression is higher in non-ciliated cells; PTCH1 is lower in G2019S animals in parallel with fewer cilia; most importantly NRTN is lower in G2019S cells in parallel with fewer cilia and lower PTCH1 and usually cilia dependent; cilia effects are also seen in human PV neurons. We modified the language throughout to avoid stating causation instead of correlation.

Minor:

1. Fig. 1: There is no explanation of why the authors check for somatostatin interneurons as well. They needed to be checked and can be studied elsewhere.

2. p.4 The description of Fig. 2B says "higher levels of LRRK2 enzyme", but it should read "higher numbers of Lrrk2 dots", since the authors did not confirm the change in protein expression or the enzymatic activity. Thank you—we corrected the text accordingly

October 21, 2024

RE: Life Science Alliance Manuscript #LSA-2024-02922-TR

Dr. Suzanne R. Pfeffer
Stanford University
Dept. of Biochemistry
School of Medicine
Stanford, 279 Campus Drive B400 94305-5307

Dear Dr. Pfeffer,

Thank you for submitting your revised manuscript entitled "Pathogenic LRRK2 mutations cause loss of primary cilia and NRTN in striatal Parvalbumin interneurons". We would be happy to publish your paper in Life Science Alliance pending final revisions necessary to meet our formatting guidelines.

- please be sure that the authorship listing and order is correct
- please use the [10 author names, et al.] format in your references (i.e. limit the author names to the first 10)
- please add a figure callout for Figure 3A and Figure 4C to the main manuscript text
- in the Materials and Methods section, please indicate the source of patient samples and that written informed consent was obtained for their use

LSA now encourages authors to provide a 30-60 second video where the study is briefly explained. We will use these videos on social media to promote the published paper and the presenting author (for examples, see <https://docs.google.com/document/d/1-UWCfbE4pGcDdcgzcmiuJI2XMBJnxKYeqRvLLrLS08s/edit?usp=sharing>). Corresponding or first-authors are welcome to submit the video. Please submit only one video per manuscript. The video can be emailed to contact@life-science-alliance.org

A. FINAL FILES:

B. MANUSCRIPT ORGANIZATION AND FORMATTING:

**Submission of a paper that does not conform to Life Science Alliance guidelines will delay the acceptance of your

manuscript.**

The license to publish form must be signed before your manuscript can be sent to production. A link to the electronic license to publish form will be available to the corresponding author only. Please take a moment to check your funder requirements.

Sincerely,

October 28, 2024

RE: Life Science Alliance Manuscript #LSA-2024-02922-TRR

Dr. Suzanne R. Pfeffer
Stanford University
Dept. of Biochemistry
School of Medicine
Stanford, 279 Campus Drive B400 94305-5307

Dear Dr. Pfeffer,

Thank you for submitting your Research Article entitled "Pathogenic LRRK2 mutations cause loss of primary cilia and NRTN in striatal Parvalbumin interneurons". It is a pleasure to let you know that your manuscript is now accepted for publication in Life Science Alliance. Congratulations on this interesting work.

DISTRIBUTION OF MATERIALS:

Again, congratulations on a very nice paper. I hope you found the review process to be constructive and are pleased with how the manuscript was handled editorially. We look forward to future exciting submissions from your lab.

Sincerely,
